# Characterization of Volcano-Sedimentary Rocks and Related Scraps for Design of Sustainable Materials

**DOI:** 10.3390/ma16093408

**Published:** 2023-04-27

**Authors:** Luisa Barbieri, Fabiana Altimari, Fernanda Andreola, Bruno Maggi, Isabella Lancellotti

**Affiliations:** 1Department of Engineering “Enzo Ferrari”, University of Modena and Reggio Emilia, Via P. Vivarelli 10, 41125 Modena, Italy; luisa.barbieri@unimore.it (L.B.);; 2CRICT—Inter-Departmental Research and Innovation Center on Constructions and Environmental Services, Via P. Vivarelli 10, 41125 Modena, Italy; 3Europomice SRL, Via N. Torriani 1, 20124 Milan, Italy

**Keywords:** volcanic rocks, volcanic scraps, materials

## Abstract

This work started as a joint academia and company research project with the aim of finding new applications for domestically sourced volcanic products and related waste (pumice, lapillus, zeolitic tuff and volcanic debris from Tessennano and Arlena quarry) by creating a database of secondary volcanic raw materials and their intrinsic characteristics to help industry replace virgin materials and enhance circularity. In this context, accurate chemical, mineralogical, morphological, granulometric and thermal characterizations were performed. Based on the results presented, it can be concluded that due to their lightness, these materials can be used in the design and preparation of lightweight aggregates for agronomic purposes or in the construction field. Furthermore, due to their aluminosilicate nature and amorphous fraction, pumice and lapillus can play the role of precursor or activator for geopolymer preparation. With its porous nature, zeolitic tuff can be exploited for flue gas treatment. Due to the presence of feldspathic phase (sanidine), these materials can be used in tile production as a fluxing component, and with their pozzolanic activity and calcium content, they have application in the binder field as supplementary cementitious material or as aggregates.

## 1. Introduction

Volcanic products are available in considerable quantities in many regions of the Earth and constitute a valuable natural and, in most cases, ecological resource, but are not fully exploited despite their enormous potential.

These raw materials are aluminosilicate-based, and their main field of application is horticulture. Volcano-sedimentary rocks can be, although to a lesser degree than in horticulture, applied in a variety of fields, such as the production of concretes, ceramic materials, insulators, adsorbents, etc. Many applications are reported in the literature, depending on the type and chemical composition of the specific material [1,2,3]. This study focuses on pumice, lapillus, zeolitic tuff, Tessennano and Arlena debris extracted from various quarry sites in Italy.

Pumice is a naturally expanded magmatic alveolar rock that originates from explosive volcanic eruptions following a violent expansion of dissolved gas in lavas of acid chemical composition. The rapid cooling of the rock prevents it from crystallizing, causing the gases dissolved within it to expand and generate a glassy foam. The inside of the rock is in fact composed of a huge number of small to medium diameter channels, interconnected with each other and with the outside [4]. This process forms a natural and environmentally friendly product, which is very dimpled; therefore, it is light, with high porosity, high water retention, slow liquid release and high thermal and acoustic insulating power.

The properties of pumice have been known for a long time; in fact, there are many studies in the literature on the use of pumice in various areas. In horticultural applications, pumice can be used as an inorganic component in substrates for green roofs, together with various organic compounds, thereby improving the chemical and physical properties of the soil, plant growth and nutrient supply and decreasing the stress to which plants are subjected [5,6]. In the same field, other authors studied the possibility for pumice to be engineered and used in the structure of a fertilizer [7], and Kong et al. investigated its use as a soil improver [8]. Pumice is also highly valued in the building industry, as it is used in concrete and mortar to achieve thermal insulation, fire resistance, freeze/thaw resistance, chemical resistance, reduction in the expansion of the alkali–silica reaction and lightness of the matrix [9]. More recently it has been studied for the production of alkali-activated materials [10,11,12]. Other fields of application for pumice can also be found in the literature, such as use as a biofilter for the removal of nitrogen from a recirculating aquaculture system [13] or as a combined medium for the purification of domestic wastewater [14,15].

Volcanic lapillus is an effusive magmatic rock (Pleistocene Vulsini Volcanite) that originates from explosive volcanic eruptions following a violent expansion of dissolved gases in lavas of acid chemical composition, as with the formation of pumice. Volcanic lapillus forms from magma with a lower silica content than that which forms pumice. The lower viscosity and slower cooling of the lava facilitates the escape of a certain amount of gas from the magma. This leads to the formation of rocks characterized by voids with a larger average diameter than those of pumice, but they are far fewer in number. It is an inert natural volcanic, naturally calcined at high temperature, porous, insulating, light and free of toxic and hazardous substances.

The properties of lapillus, compared with pumice, are less well known, but it is a suitable material in technical green areas for creating drainage, and is ideal for lawns and sports facilities. It is also used in construction as a thermal and acoustic insulator and for rehabilitating masonry and reinforced concrete works [16], and some authors have studied its alkaline activation for the manufacture of geopolymers [17].

Zeolitic tuff is a volcanic product with high and selective cationic exchange power and has the capacity to neutralize, by retaining them in its structure, leachable elements such as ammonium, heavy metals and organic molecules, and to absorb odorous gases such as ammonia, hydrogen sulfide and mercaptans.

Among the materials mentioned so far, zeolites and zeolitic products are the most studied; they are of great use in the agricultural field as they improve the physical characteristics of soil and make it possible to reduce the use of synthetic fertilisers as well as irrigation water. They minimize NO_3_-leaching from soil and optimize crop growth and yield. In fact, numerous applications can be found, for example, in wheat, rice crops and maize [18,19,20], and in the oenological field [21]. They have also been investigated as an adjunct to fertilisers [22,23]. Due to their high adsorbing and filtering power, they can also be used for decontaminating heavy metals [24,25] and for removing harmful elements from wastewater [26]. In addition, there are examples of their use in construction for the manufacture of high-performance self-hardening concretes [27] and in cementitious materials [28], where the addition of zeolite improves certain properties.

In this paper, the rocks and debris mentioned were subjected to chemical-physical, mineralogical, thermal and microstructural characterisation, and compared in order to investigate their potential application (alone or mixed with each other or mixed with other raw materials) in tailoring of new products useful in different fields, including construction, agronomy and industrial emission management. In the literature, there are no papers simultaneously highlighting and comparing the physical-chemical-thermal-morphological properties of these natural minerals (and their possible extraction scraps). Therefore, the aim of the paper is to create a database of secondary volcanic raw materials and their intrinsic characteristics to help industry use them to replace virgin materials and, thereby, enhance circularity.

## 2. Materials and Methods

### 2.1. Volcanic Products

Volcanic rocks are extracted from quarries in the territory between the Vulsino Volcanic District (extending from southern Tuscany to northern Lazio around Lake Bolsena) and the Cimino–Vicano Volcanic District (around Lake Vico, further south than the previous one).

Latium volcanoes belong to two clearly different magmatic series. The first includes the acidic, rhyolitic and rhyodacites volcanism of the Cimini, Tolfa and Ceriti Mountains and is older than the second series which includes the Vulsino, Vicano, Sabatino and Colli Albani groups, and shows a strongly alkaline-potassium nature. Latium, therefore, represents the junction area of these two different systems [29].

The pumice, lapillus, zeolitic tuff and debris from the Tessennano and Arlena quarries used in this work were supplied by Europomice srl. Pumice is mined in Tuscany, in Pitigliano (GR) at ‘Poggio Nardeci’, and in Latium, in Tessennano (VT) at ‘Riserva Muraccio’ and Arlena di Castro (VT). Lapillus and zeolitic tuff are mined in Latium in Cellere (VT) at “Monte Cellere” and in Tessennano (VT) at “Riserva Muraccio”, respectively. Tessennano and Arlena debris are mined in quarries in northern Latium. Arlena and Tessennano sands are the scrap from the processing of pumice, hereafter called sands for simplicity.

Figure 1 shows the Pitigliano pumice quarry (Figure 1a) and a sample (Figure 1b) characterized by a matrix within which are embedded pumiceous clasts that exhibit gray coloration. The sample also has elongated pseudo rectangular transparent crystals, probably potassium feldspar. Figure 2 shows a lapillus sample (Figure 2a) and zeolitic tuff (Figure 2b).

All the materials used in this study, except the zeolitic tuff, are quarry scraps.

### 2.2. Chemical and Physical Characterization

Approximately 1000 g of a well-mixed sample was taken from each mineral in order to ensure a representative sample of the entire bank of material. The samples were subjected to particle size analysis, pH and specific conductivity measurements, true density, chemical and mineralogical analysis, and finally investigated with a scanning electron microscope (SEM). Before performing the analyses, the samples were dried in an oven for 24 h at a temperature of 100 ± 5 °C.

The particle size analysis was performed by taking 100 g of sample and sieving it first manually, using stacked sieves with an average hole diameter of 2 mm and 1 mm, to separate the coarse fraction from the fine one. The analysis was then performed with a laser particle size analyzer (Malvern Instruments (Malvern, UK), model Hydro200S).

In order to evaluate the chemical stability of the samples by assessing the release of ions into the aqueous environment, pH and specific conductivity measurements were carried out using a portable pH tester (Laboratory PH sensor Hamilton type Liq-glass SL, OAKTON Eutech Instruments (Vernon Hills, IL, USA) pH 5/6 and Ion 6) and a portable conductivity tester (OAKTON Eutech Instruments CON6/TDS 6, Vernon Hills, IL, USA), respectively. A few grams of sample were taken and placed inside a beaker containing bi-distilled water with a sample–water ratio of 1:10. The test was performed in dynamic mode (700 rpms) for 8 h; measurements were taken at predetermined times, i.e., at the initial instant and after 5, 15, 30, 60, 120, 240 and 480 min. The above reported procedure is in accordance with UNI EN 13037:2012 (pH rule standard) and UNI EN 13038:2012 (conductivity rule standard).

True density (ρ_abs_) measurements were assessed for each mineral with a helium pycnometer (Mycromeritics Accupyc 1340, Norcross, GA, USA) by taking a few grams of sample and placing them inside a 10 cm^3^ measuring cell. Bulk density (ρ_bulk_) was measured by Geopyc 1360 Envelope Density Analyzer (Micromeritics, Norcross, GA, USA). In this analysis, the sample is placed in a cylindrical chamber equipped with a piston. The measure is run before with the empty chamber to determine the total volume and after with the granular sample. The volume of the sample is determined by subtracting the volume of the empty chamber from the volume of the same chamber after the sample has been added.

The chemical analysis was performed by sequential X-ray Fluorescence Spectrometer (XRF) using an ARL PERFORM’X instrument, and the mineralogical analysis was performed by X-ray diffraction with the Rietveld method using an X’PERT 3 POWDER PANALYTICAL instrument. The phase fractions were extracted by the Rietveld refinements, using GSAS software and they were rescaled on the basis of the absolute mass of corundum originally added to the mixtures as an internal standard. The chemical and mineralogical analyses were provided by the volcanic raw material company.

Microstructural analysis was conducted using an environmental scanning electron microscope (ESEM), model ESEM-Quanta 200, coupled with an X-ray EDS microanalysis system, model X-EDS Oxford INCA-350, to perform the chemical analysis (on lapillus and pumice only).

### 2.3. Thermal Characterization

In order to investigate the possible fields of use of the materials, their thermal transformations were studied by thermo-gravimetric and differential thermal analyses (TGA/DTA) and hot stage microscopy (HSM).

The samples were appropriately ground with an agate mortar to avoid possible contamination and powdered, reaching an average particle diameter of less than 20 μm. The samples were analyzed with a heating rate of 10 °C/min.

DTA allows determination of the thermal reactivity of the powders as the temperature increases, while TGA determines the change in mass as a function of temperature resulting from the development or absorption of gases. The TGA/DTA analyses were performed simultaneously with the NETZSCH STA 429 (CD) instrumentation.

The HSM analysis was performed with Expert System Solutions instrumentation, model M3MD1600/80/2. The samples were first reduced to powder and then pressed to form very small cylinders 2 mm in diameter and 3 mm in height. Images were taken at a heating rate ramp of 10 °C/min until 1300 °C. During the post-analysis phase, by means of calculations on the shape of the specimen, the instrument software is able to attribute the characteristic temperatures of the transformation states (sintering, softening, sphere, hemisphere and melting temperatures) for each sample analyzed.

## 3. Results

### 3.1. Chemical and Physical Characterization

#### 3.1.1. Particle Grain Size Analysis

The data obtained for the sample fraction less than 1 mm are represented by two curves on a semi-logarithmic scale: a cumulative curve (whose corresponding y-axis is on the right-hand side), and a bimodal distribution curve. The characteristic diameters are also shown.

Lapillus presented the following particle size distribution: 6.5% of particles had diameters greater than 2 mm, 17% had diameters between 1 mm and 2 mm, and the remaining 76.5% had diameters of less than 1 mm. The fraction with a diameter of less than 1 mm is visible in Figure 3. Based on the reported data, 90% of the distribution is under 65 μm and 100% is under 753 μm. The D50 value indicates that the 50% of the distribution was under 22 μm.

Pumice presented the following particle size distribution: 27% of particles had diameters greater than 2 mm, 31% had diameters between 1 mm and 2 mm, and the remaining 42% had diameters of less than 1 mm. The fraction with a diameter of less than 1 mm is visible in Figure 4. Based on the data reported, 90% of the distribution is under 278 μm and 100% is under 851 μm. The contrasts with the distribution for lapillus, for which only 81% of the sample was under 65 μm. The D50 value indicates that the 50% of the distribution was under 22 μm.

Arlena sand presented the following particle size distribution: 15% of particles had a diameter greater than 2 mm, 15.5% had a diameter between 1 mm and 2 mm, and the remaining 69.5% had a diameter of less than 1 mm. The fraction with a diameter of less than 1 mm is visible in Figure 5 where the 50% of the distribution is less than 30 μm.

Tessennano sand presented the following particle size distribution: 33% of particles had a diameter greater than 2 mm, 23% had a diameter between 1 mm and 2 mm, and the remaining 44% had a diameter of less than 1 mm. The fraction with a diameter of less than 1 mm is visible in Figure 6. The D90 value was 561 μm, and the D50 was 76 μm.

The particle size distribution of the sands indicates that the material is coarser than lapillus and pumice.

The particle size analysis was not performed on zeolitic tuff because the grain size depends on its use, so the material is ground by the company to the required size. The sample extracted from the quarry presented a large grain size range (from 7–35 mm up to 1–100 μm).

#### 3.1.2. Chemical and Mineralogical Analyses

The results of the XRF analysis shown in Figure 7 indicate that the materials have an aluminosilicate character.

The highest SiO_2_ content was observed in the two sands (around 60%).

Fe_2_O_3_, CaO, MgO, K_2_O and Na_2_O were detected in all materials. Lapillus had a higher Fe_2_O_3_ and CaO content than the other materials. With the exception of lapillus, in all the other volcanic products, K_2_O varied between about 5.2% and 8.5%. The two sands presented a very similar chemical composition. In the zeolitic tuff, the content of Na_2_O was very low. Loss of ignition of zeolitic tuff is higher than for the other materials due to its characteristic content of water. The chemical data are in line with the literature for pumice, with SiO_2_ and Al_2_O_3_ levels of 48% and 14.9%, respectively. Moreover, an appreciable iron oxide content (6.71%) was observed. Other oxides, such as MgO, Na_2_O, CaO and K_2_O, were also detected [30].

The mineralogical analysis (Table 1) showed that all the products under investigation exhibited an amorphous fraction; this was very low in zeolitic tuff and lapillus and very high in pumice and sands. Zeolitic tuff was similar to pumice in terms of the amount of quartz detected. It is close to lapillus in terms of the amount of amorphous fraction detected and, like the two sands, showed a small percentage of biotite. Sanidine and anorthite were detected in pumice, lapillus, and in Arlena and Tessennano sands. Muscovite and phlogopite were only detected in the pumice, while hematite, plagioclase and mica only in the lapillus. Tessennano sand was the only material analyzed that did not contain analcime. Arlena sand and zeolitic tuff showed 9% and 54% of chabazite, respectively.

#### 3.1.3. pH, Specific Conductivity and Density

The pH and specific conductivity were measured under stirring conditions and their trends are reported in Figure 8 and Figure 9. Based on these plots, it is evident that the pH values were all very similar and near to neutrality. On the other hand, the conductivity showed more significant differences among the different volcanic materials and an increasing trend with time.

The true and bulk density values for each material can be seen in Table 2.

True density corresponds to the mass of a particle divided by its volume, excluding open and closed pores. Generally, this kind of density depends on the chemical compositions of the volcanic products. Zeolitic tuff showed a lower true density value than the other products, while the two sands presented very similar values. Lapillus showed a higher density value than the other volcanic rocks while pumice showed an intermediate value.

The bulk density of a powder is the ratio of the mass of an untapped powder sample and its volume including the contribution of the inter particulate void volume. Hence, the bulk density depends on both the density of powder particles and the spatial arrangement of particles in the powder bed.

The bulk density values measured on volcanic products show that pumice is the lighter material. This indicates the presence of intraparticle and interparticle porosity; therefore, lower values were observed for more porous and lighter materials, such as pumice, lapillus and zeolitic tuff, with respect to the sands.

#### 3.1.4. SEM Analysis

The SEM analysis was performed on the pumice and lapillus samples only.

The microstructural analysis of lapillus sample (Figure 10) showed:A wide grain size range between 20 and 400 μm;Low mutual densification between particles;Particles with irregular shapes and jagged edges;High interparticle porosity regardless of particle size.
Figure 10Lapillus: micrographs at 300× (**a**) and 1200× (**b**) in backscattered electron (BSE) mode.
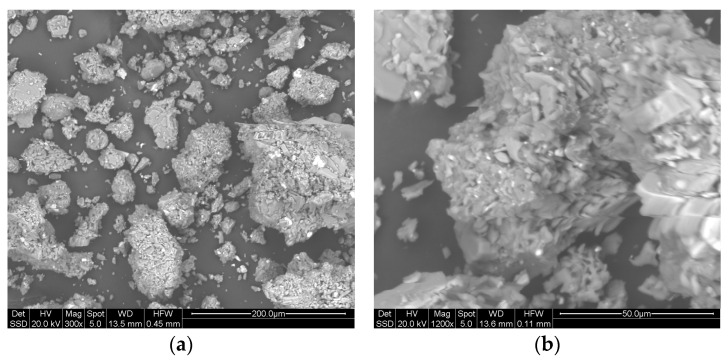



The microstructural analysis of the pumice sample (Figure 11) showed:A high proportion of particles with sizes in the range 100–200 micron;Particles with a more regular shape than those of the lapillus sample;Greater interparticle thickening;Presence of non-porous glassy zones (circled in Figure 11b) corresponding to the vitreous fraction detected by XRD (79.7%).
Figure 11Pumice: micrographs at 300× (**a**) and 1200× (**b**) in BSE mode.
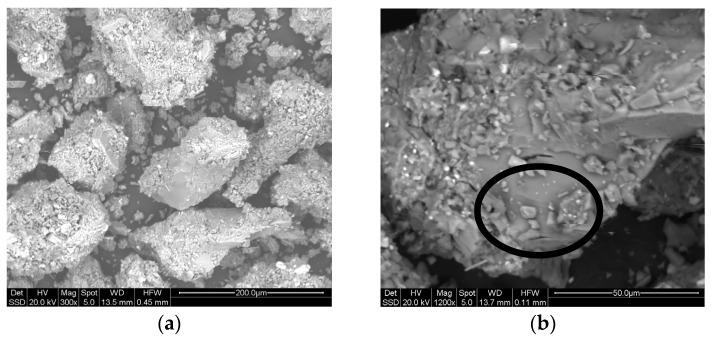



Based on the chemical analysis described above, the total elements were determined, while with the SEM together with the EDS probe, it was possible to analyze very small crystalline zones and the amorphous areas to assess the possibility of alkaline activation of the materials.

The semi-quantitative chemical analysis performed with EDS is reported in Table 3.

### 3.2. Thermal Characterization

#### 3.2.1. Thermogravimetric and Differential Thermal Analysis (TGA/DTA)

The thermal investigation was conducted on all the raw materials with the exception of zeolitic tuff, as it could not be easily subjected to thermal processes.

The following figures describe the thermograms with their peaks (endothermic or exothermic) and the weight variation of the material under study. The red curves describe the TG while blue curves describe the DTA.

Figure 12 shows the results of the analysis conducted on lapillus.

The red curve showed a material with a low LOI (loss on ignition) of around 1%, indicating its high thermal stability; the sample has a low interstitial water content.

Three main events can be described from the blue curve:Up to a temperature of about 200 °C, there was a loss of moisture resulting in a loss of about 0.5% of the initial weight;Between 500 °C and 1000 °C, substantial stability of the material was noted during heating;Around 1200 °C, there was an endothermic peak due to the melting of the crystalline lattice constituting the material, which as confirmed by the mineralogical analysis data, has an exclusively crystalline microstructure (approx. 87%).

The behavior of pumice with increasing temperature can be seen in Figure 13.

Compared with lapillus, pumice showed a greater weight reduction. A LOI of around 6% was observed in the TG analysis, which is in agreement with other authors [30,31], with the greatest weight loss at temperatures below 500 °C due to the removal of moisture as well as OH species linked to metal oxides. With reference to DTA, an endothermic event was observed around 450 °C, which could be traced back to the dihydroxylation of the muscovite clay. The material remained stable up to around 1000 °C and then underwent melting of different crystalline phases at temperatures of around 1050 °C and 1200 °C, resulting in two separate peaks.

The behavior of Arlena sand can be observed in Figure 14.

The blue curve showed a peak at a temperature below 200 °C, which is attributable to the loss of moisture, and a peak at around 1160 °C due to the melting of the crystalline phases.

The red curve showed a total weight loss of 8%, although this phenomenon mainly occurred at T < 400 °C.

The behavior of Tessennano sand can be observed in Figure 15.

The TG showed a weight loss of about 8%, which occurred mainly at T < 600 °C.

The DTA allowed three main events to be described:Up to a temperature of about 200 °C, there was a loss of moisture;Between 400 °C and 500 °C, there was a loss of reticular water most probably related to biotite;Around 1200 °C, there was an endothermic peak due to the melting of the crystalline phases.

#### 3.2.2. Hot Stage Microscopy

Hot stage microscopy allowed the characteristic temperatures of the samples to be determined (Table 4). This knowledge is helpful in determining the optimum ranges of manufacturing conditions for ceramic processes.

The HSM provides information about the characteristic temperatures at which changes occur:Sintering temperature: the temperature at which the sample is reduced by about 5% and the sintering process of the grains begins;Softening temperature: the temperature at which the sample takes on a plastic character, and both the upper profile and the edges tend to round off;Sphere temperature: the temperature at which the height and width have the same magnitude, resulting in a spherical shape;Semi-sphere temperature: the temperature at which the width of the sample reaches dimensions twice as large as the height;Melting temperature: the temperature at which the width of the sample is three times its height.

## 4. Discussion

In this study, the characterisation of the materials, which are almost all extraction scraps, was driven by the need to investigate their alternative uses.

Lapillus, pumice and the two types of debris (Arlena sand and Tessennano sand) were the extraction scraps for which the particle size curve was analyzed. No scraps are produced in the extraction of zeolitic tuff. Zeolitic tuff can be exploited for various uses. If it is very fine (<250 μm), it can be used for foliar treatments in organic farming and for gas purification possibly after a micronization process [32,33,34], while if it is coarse (>1 mm), it is used for wastewater treatment due to its capacity for adsorbing contaminant [35] or as a soil conditioner for improving physical and chemical soil properties [36].

The chemical and mineralogical analyses showed that all the materials are aluminosilicates. The two volcanic debris were very similar to each other in both chemical and mineralogical composition and to pumice because they are scraps coming from the extraction operations of pumice. They also show a high fraction of vitreous phase (around 60–70%) due to their volcanic origin. Lapillus and pumice, although both materials originating from explosive type eruptions from magmas of acid composition, show differences that are probably due to the different rates of solidification. Lapillus has much higher amounts of Fe and Ca, resulting in its darker color, while pumice has higher amounts of Na and K. Both materials were observed to contain amounts of Si and Al comparable to the values reported in the literature [9]. Pumice presented the highest amount of amorphous phase (around 80%), and compared with lapillus, had higher alkaline oxides (Na_2_O + K_2_O) and lower alkaline-earth oxides (CaO + MgO) contents, with a corresponding melting role in high temperature applications. This is also supported by lower sintering and softening temperatures detected by hot stage microscopy.

The pH and specific conductivity were assessed for all volcanic products in stirred mode to understand the tendency of the materials to release ions in solution. The pH values varied between 6 and 7.6, and being within the neutrality, the products are suitable for agronomic purposes [37]. Concerning electrical conductivity, lapillus was the most stable in terms of ions released in solution, compared with the other volcanic products. This behavior can be attributed to the lower amount of potassium (3.66%), a highly soluble ion, in lapillus. Pumice released more ions in water and showed an increasing tendency related to the higher amount of K^+^ ions (8.55%). The two sands showed almost similar values, especially under stirring conditions.

As far as true density is concerned, the reported values allow some conclusions to be drawn: The higher values for lapillus can be related to the presence of higher density crystalline phases such as anorthite (2500–2800 Kg/m^3^) and hematite (5120–5300 Kg/m^3^). The values for pumice can be related to the presence of crystalline phases such as sanidine (2550–2630 Kg/m^3^). The sand values are mainly attributable to the density values of quartz (2650 Kg/m^3^) and sanidine, and the lower values of zeolitic tuff are attributable to the presence of chabazite.

The bulk density values presented demonstrate the lightness of these materials. For this reason, they can be exploited in the formulation of lightweight aggregates. In particular, the lower bulk density value in pumice results in better behavior in lightweight aggregates, as already observed by Righi et al. [7].

SEM analysis was conducted on pumice and lapillus only. The SEM analysis agrees with the mineralogical analysis, confirming that lapillus is more crystalline while pumice is more amorphous, as already reported by Pinarci et al. [38]. Given the aluminosilicate nature of both these materials with a significant amorphous fraction, they have been used by other researchers as precursors to obtain geopolymers [10,17]. In the literature, no uses of lapillus are found. However, Szabo et al. produced geopolymers starting from mechanically activated pumice [39], and Ulusu et al. produced pozzolanic cement with a high volume of pumice [40].

Furthermore, due to their amorphous silica content these products could be used also as precursors for obtaining activating solutions. Indeed, other studies in the literature have successfully exploited other similar materials, such as diatomaceous earth, to activate aluminosilicate raw materials. Diatomite is a sedimentary rock with a high amorphous silica content similar to the volcanic debris discussed here, and it is used as silica source for obtaining waterglass to activate aluminosilicate precursors [41]. The EDS analysis confirmed the mineralogical characterization showing that the crystalline part of pumice is enriched in Si, Al and K, corresponding to the sanidine crystalline phase, while in the amorphous fraction, alkaline earth elements (Ca and Mg) are concentrated together with Si and Al. Furthermore, the EDS analysis showed an Si/Al ratio of 2.29 for the crystalline fraction and a ratio of 2.52 for the amorphous fraction. The latter is particularly reactive in an alkaline environment. The Si/Al ratio is very important in the design of a geopolymer because it influences the physical properties of the material: if it is between 2 and 3, a structure with rigid bonds is obtained and the material is comparable to cements and ceramics [42,43]. The importance of amorphism in an aluminosilicate precursor for geopolymers is confirmed in the literature [44]. Finally, thermal analyses were conducted to also assess the behavior of materials when subjected to firing, as in the case of lightweight aggregate or ceramic tile production.

It can be concluded that, with the exception of zeolitic tuff, these materials could be exploited in aluminosilicate matrices as substitutes for virgin raw materials, both for cold and hot consolidation. This is important since, given the current geopolitical situation, some raw materials are difficult to obtain, and these materials could replace, for example, feldspars or fluxes in ceramic pastes and glaze compositions, clays in lightweight aggregates (LAWs) or metakaolin in geopolymers.

In a previous study [17], the authors used lapillus and pumice to completely replace the red clay in LWAs compositions, using spent coffee ground (SCG) for pore-forming, and employing the same optimized ratio for clay (volcanic minerals/porous agent 85/15 wt%/wt%). Two compositions were prepared: one with only pumice + SCG (85 wt% + 15 wt%), and the other with a mix of equal parts of pumice and lapillus + SCG (85 wt% + 15 wt%). The lapillus-based compositions were not fit for purpose. The spherical samples were fired at 1000 °C for 1 h in an electrical kiln that was already hot in order to simulate the thermal shock that the aggregates undergo during industrial processes, which however, generally occur at higher temperatures (from 1200 to 1400 °C). The preliminary results obtained highlighted that the tailored compositions complied with the standard regarding the required particle density being lower than 2000 Kg/m^3^ (UNI EN 13055:2016 rule). In addition, they had a total porosity of around 60%, which is similar to values found in commercial products such as Arlita Leca L. These physical features suggest their application as a draining medium on green roofs. In particular, low density is an important factor when choosing the drainage components in the layers. In Figure 16, it is interesting to note that the substitution of clay by volcanic scraps has a positive effect on the density. This valorization of volcanic scraps represents a valuable way to manufacture light-weight aggregates for agronomic purposes on green roofs. This could be significant considering the advantages of green roofs for reducing the urban heat island effect, the energy consumption of buildings, air and noise pollution, as well as improving the quality and regulation of rainwater, and protecting biodiversity.

Pumice and lapillus could also be used for the production of cement mortars or lightened concretes, both as supplementary cementitious materials, i.e., by substituting Portland Cement, or by substituting commercial aggregates. In addition, Arlena and Tessennano sands could replace commercial sands in the construction industry. Preliminary laboratory tests have shown the possibility of incorporating them as an inert fraction in a geopolymer mortar, which by their mechanical performance (their measured compressive strength was about 2.6 N/mm^2^) fall within the class of performance-guaranteed mortars according to UNI EN 998-2.

The two sands could be exploited as an alternative inorganic component for the formulation of alternative substrates to peat. Further, both sands and zeolitic tuff, could be used in the design of technical green areas.

Zeolitic tuff and pumice could be used for flue gas treatment and water purification. In particular, regarding the use of zeolitic tuff for gas purification, a previous study examined the design of a plant with a process that uses natural chabazite-based zeolitic tuff powder to purify industrial gases from harmful organic substances. The purification of the gas takes place thanks to the open structures with high surface areas typical of zeolitic tuff. This plant could replace existing combustion-type plants, which have the problem of not being able to remove particularly low concentrations of pollutants. It could also replace activated carbon filters, whose efficiency is conditioned by a series of parameters such as the molecular weight and concentration of pollutants, temperature, humidity, pressure [45].

The potential of these materials is very high. The latest studies performed by the authors focus on the replacement of foreign feldspar by Italian pumice or lapillus scraps as fluxing agents in the production of ceramic bodies. They are also able to confer color, thereby reducing the use of dye pigments, thanks to chromophore oxides present within the pumice and lapillus.

In the future, it would be interesting to compare the behavior of these volcanic rocks with sedimentary rocks with similar properties. For instance, diatomaceous earths are silicate raw materials with low density, high porosity, good absorbability, and other characteristics similar to pumice or zeolitic tuff. However, the different origin and texture could lead to different reactivity. The vesicular rough-textured nature of volcanic glass, which may or may not contain crystals of pumice, should give it greater responsiveness and efficiency in certain contexts compared with diatomaceous earths constituted by fossilized diatom skeletons. Some authors have already experimented with the possibility of obtaining porous materials for horticulture purposes with long-term nutrient effect based on a vitrifying agent in the form of pumice scraps [7].

Moreover, in the field of abatement of pollutants (in particular, pollutants of organic nature, such as C_6_H_6_) in industrial gaseous streams, the efficiency shown by zeolitic tuff, thanks to its open structures with high surface areas and ionic exchange capability, is practically unique and is subject to a patent by the authors [45].

## 5. Conclusions

In this paper, we analyze and report on the main work performed on volcanic rocks and materials derived from them, with the aim to create a database from which industry can take inspiration for enhancing circularity by using these raw materials in production cycles.

From the results presented in this paper we conclude that:Due to the lightness of the volcanic products, they can be used in the design and preparation of lightweight aggregates useful for agronomic purposes or in the construction field;Due to their aluminosilicate nature together with the presence of an amorphous fraction, pumice and lapillus can play the role of precursor for geopolymer preparation;Zeolitic tuff can be exploited for flue gas treatment, which is made possible by its porous nature and open structures with high surface areas;Due to the presence of feldspathic phase (sanidine), volcanic debris can be used in tile production as the melting component. Thanks to its pozzolanic activity and calcium content it could also be used in binders as supplementary cementitious material or as aggregate.

## Figures and Tables

**Figure 1 materials-16-03408-f001:**
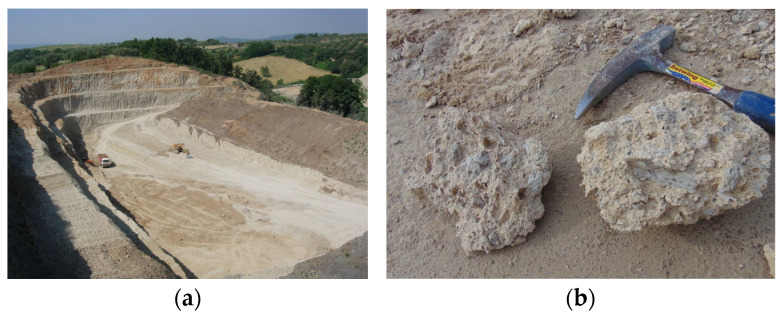
Pitigliano pumice quarry (Grosseto, Tuscany) on the **left** (**a**) and pumice stone on the **right** (**b**) [16].

**Figure 2 materials-16-03408-f002:**
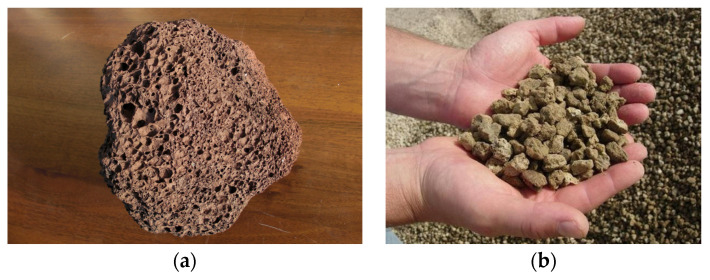
Lapillus (**a**) and zeolitic tuff (**b**) [16].

**Figure 3 materials-16-03408-f003:**
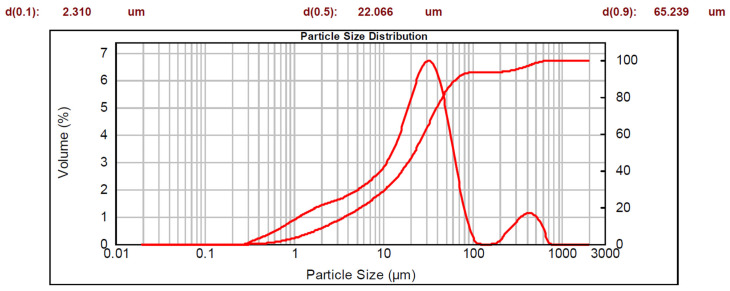
Particle size distribution of lapillus.

**Figure 4 materials-16-03408-f004:**
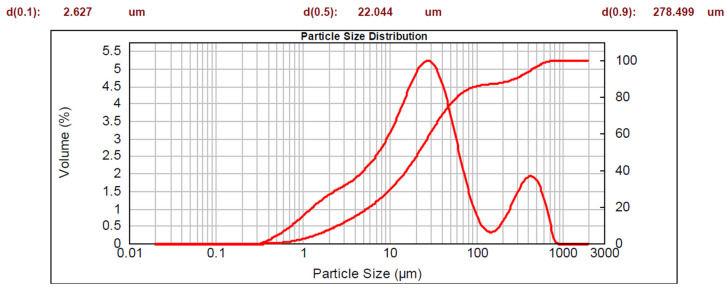
Particle size distribution of pumice.

**Figure 5 materials-16-03408-f005:**
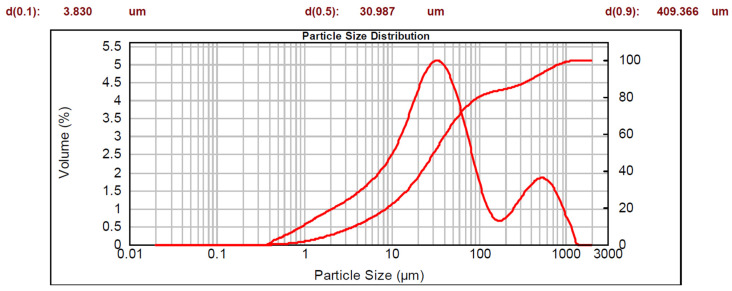
Particle size distribution of Arlena sand.

**Figure 6 materials-16-03408-f006:**
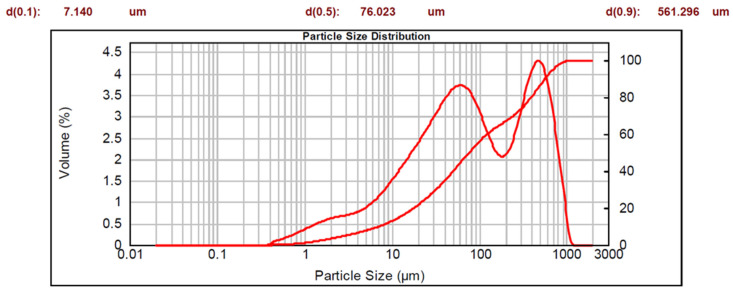
Particle size distribution of Tessennano sand.

**Figure 7 materials-16-03408-f007:**
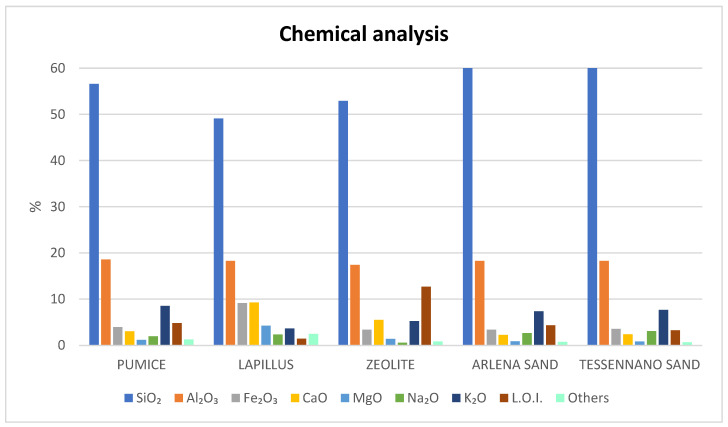
Chemical analysis (XRF) in wt% oxide of all the volcanic products.

**Figure 8 materials-16-03408-f008:**
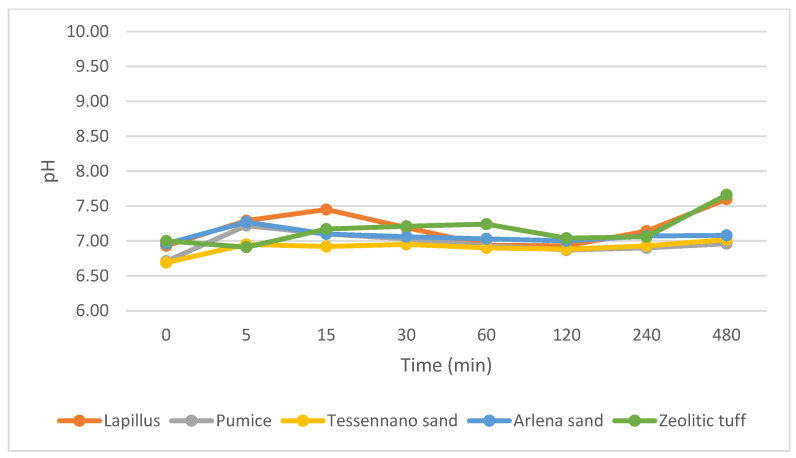
pH under stirring condition.

**Figure 9 materials-16-03408-f009:**
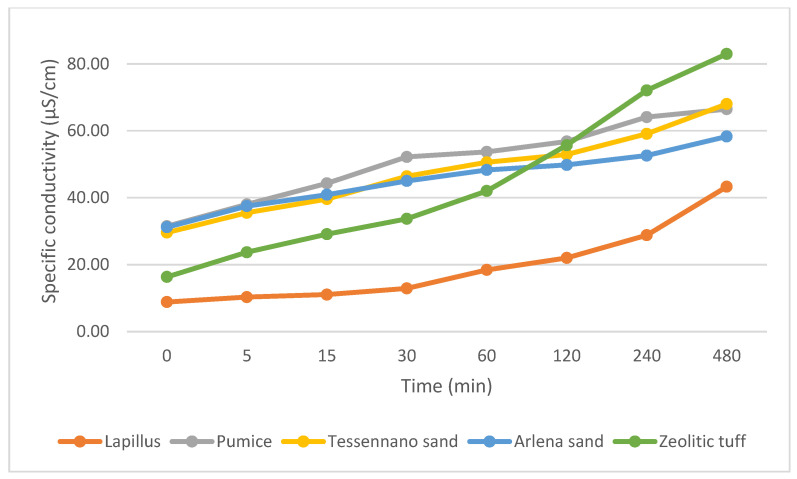
Specific conductivity under stirring condition.

**Figure 12 materials-16-03408-f012:**
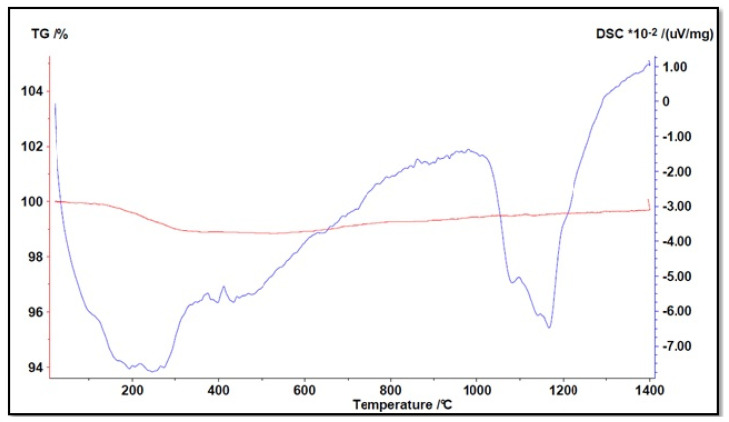
Simultaneous TGA/DTA analysis of lapillus (↑ exo peaks, ↓ endo peaks).

**Figure 13 materials-16-03408-f013:**
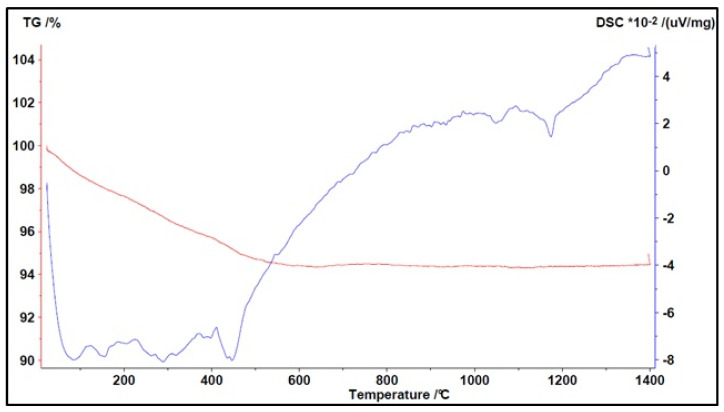
Simultaneous TGA/DTA analysis of pumice (↑ exo peaks, ↓ endo peaks).

**Figure 14 materials-16-03408-f014:**
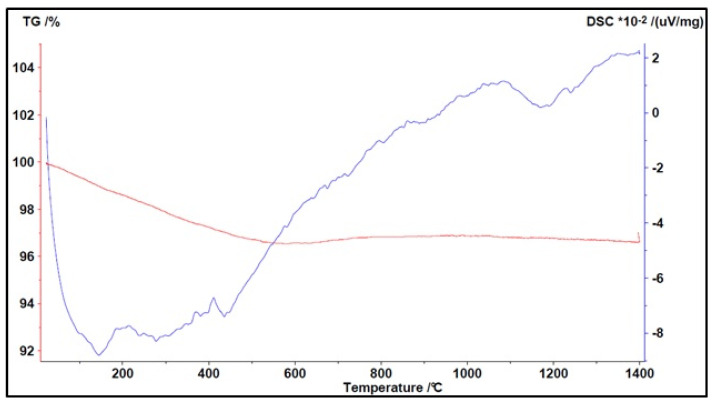
Simultaneous TGA/DTA analysis of Arlena sand (↑ exo peaks, ↓ endo peaks).

**Figure 15 materials-16-03408-f015:**
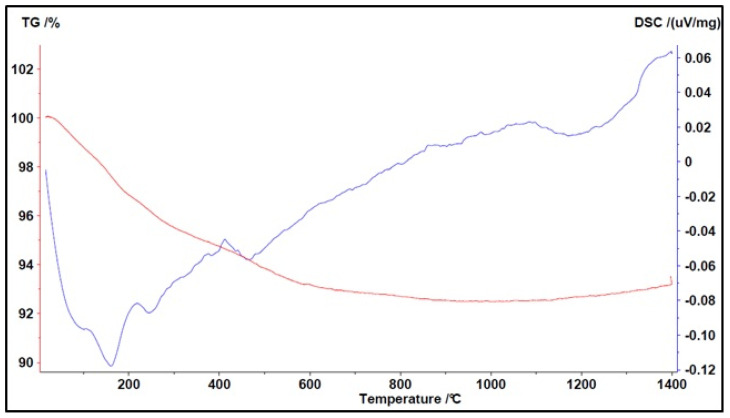
Simultaneous TGA/DTA analysis of Tessennano sand (↑ exo peaks, ↓ endo peaks).

**Figure 16 materials-16-03408-f016:**
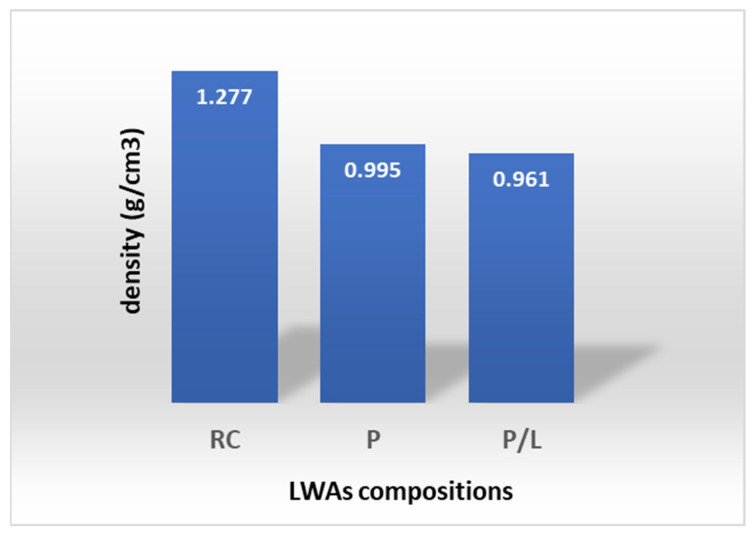
Particle density of the composition analyzed (RC only red clay; P: pumice; P/L: pumice lapillus).

**Table 1 materials-16-03408-t001:** Quantitative mineralogical analyses (wt%).

Mineralogical Phase	Lapillus	Pumice	Zeolitic Tuff	Arlena Sand	Tessennano Sand
Amorphous	16.1	79.7	11.0	61.7	78.2
Quartz (SiO_2_)	-	1.1	1.0	3.5	2.0
Sanidine (K,Na)(Si,Al)_4_O_8_	19.8	11.2	-	18.2	16.0
Anorthite (CaAl_2_Si_2_O_8_)	26.4	3.0	-	3.2	3.0
Biotite (K(Mg,Fe_2+_)_3_(AlSi_3_O_10_(OH,F)_2_)	-	-	6.0	0.5	0.8
Chabazite (Ca,Na_2_,K_2_,Mg)Al_2_Si_4_O_12_∙6H_2_O	-	-	54.0	9.4	-
Phyllipsite (Ca,Na_2_,K_2_)Al_6_Si_10_O_32_∙12H_2_O	-	-	6.0	-	-
Analcime (NaAlSi_2_O_6_∙H_2_O)	6.1	0.6	1.0	1.0	-
Pyroxenes	-	-	2.0	-	-
Feldspars	-	-	19	-	-
Diopside (CaMgSi_2_O_6_)	19.0	-	-	2.5	-
Muscovite (KAl_2_(Si_3_Al)O_10_(OH,F)_2_)	-	3.8	-	-	-
Phlogopite (KMg_3_(Si_3_Al)O_10_(F,OH)_2_)	-	0.6	-	-	-
Hematite (Fe_2_O_3_)	4.9	-	-	-	-
Plagioclase (Na,Ca)(Si,Al)_4_O_8_	5.8	-	-	-	-
Mica X_2_Y_4–6_Z_8_O_20_(OH,F)_4_	1.9	-	-	-	-

**Table 2 materials-16-03408-t002:** True and bulk density data for all the rocks analyzed.

Samples	True Density (kg/m^3^)	Bulk Density (kg/m^3^)
Lapillus	2843.8 ± 0.6	750–1150
Pumice	2579.3 ± 1.6	480–880
Zeolitic tuff	2284.3 ± 1.1	700–1000
Arlena sand	2475.0 ± 0.8	900–1100
Tessennano sand	2435.2 ± 1.0	900–1100

**Table 3 materials-16-03408-t003:** EDS analysis of crystalline and/or amorphous areas in pumice and lapillus samples.

Average Chemical Composition (EDS)	Lapillus (Crystalline Zone)	Pumice (Crystalline Zone)	Pumice (Amorphous Zone)
O	53.9%	52.6%	61.7%
Si	19.8%	21.5%	16.4%
Al	8.9%	9.4%	6.5%
Fe	5.8%	3.9%	2.7%
K	2.6%	3.7%	0.3%
Na	2.6%	2.8%	Traces
Mg	1.3%	1.6%	4.8%
Ti	0.5%	0.5%	0.4%
Ca	-	4.5%	7.2%
P	0.3%	-	-

**Table 4 materials-16-03408-t004:** Characteristic temperatures.

Samples	T Sintering (°C)	T Softening (°C)	T Sphere (°C)	T Semi-Sphere (°C)	T Melting (°C)
Lapillus	1132	1181	n.a.	n.a.	1224
Pumice	972	1004	n.a.	1274	1319
Zeolitic tuff	949	1144	n.a.	1245	1314
Arlena sand	970	1210	n.a.	1298	1341
Tessennano sand	1033	1227	n.a.	1307	1347

## Data Availability

Not applicable.

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
