# Peer review of "Characterization of Volcano-Sedimentary Rocks and Related Scraps for Design of Sustainable Materials"

_materials, 2023, doi:10.3390/ma16093408_

Round 1

Reviewer 1 Report

Well written and interesting paper. One minor comment: row 90: The authors could emphasize that the aim of this research is besides understanding also give suggestions (as done in Conclusions) of new applications for studied volcanic materials 

Author Response

Dear Editors and Reviewer of Materials

In this document, we present our response for each one of the Reviewers’ comments, as well as the list of changes and/or rebuttal made for the manuscript entitled “Characterization of volcano-sedimentary rocks and related scraps for design of sustainable materials”.

We’d like to thank the valuable comments of the Reviewers, which helped us to improve the quality of the manuscript, making it more suitable for publication in Materials than in its previous version. In the revised manuscript, all the changes have been highlighted in yellow. English language has been improved.

Reviewers' comments:

  • Reviewer #1

Well written and interesting paper. One minor comment: row 90: The authors could emphasize that the aim of this research is besides understanding also give suggestions (as done in Conclusions) of new applications for studied volcanic materials. 

Response from the authors:

We thank Reviewer 1 for her/his comments regarding our work. The text modified can be seen at the end of the Introduction section.

Reviewer 2 Report

The work performed and elaborated in this manuscript is excellent. However, I highly recommend that the manuscript is reviewed by a native speaker. 

The manuscript provides a detailed explanation (characterization) for pumice, lapillus, tuff and sand (2).  Yet, there is one point that the authors can improve. I highly recommend that in the discussion section, a better explanation is made in the sense of explaining how certain property has a positive impact thus making each of the materials suitable for production of sustainable materials. (example: compare pumice with diatomaceous earth the starting materials for production of porous materials under hydrothermal technology; see if these materials are suitable for production of geopolymers etc.). 

Author Response

Dear Editors and Reviewer of Materials

In this document, we present our response for each one of the Reviewers’ comments, as well as the list of changes and/or rebuttal made for the manuscript entitled “Characterization of volcano-sedimentary rocks and related scraps for design of sustainable materials”.

We’d like to thank the valuable comments of the Reviewers, which helped us to improve the quality of the manuscript, making it more suitable for publication in Materials than in its previous version. In the revised manuscript, all the changes have been highlighted in yellow. English language has been improved.

Reviewers' comments:

  • Reviewer #2

The work performed and elaborated in this manuscript is excellent. However, I highly recommend that the manuscript is reviewed by a native speaker. 

The manuscript provides a detailed explanation (characterization) for pumice, lapillus, tuff and sand (2).  Yet, there is one point that the authors can improve. I highly recommend that in the discussion section, a better explanation is made in the sense of explaining how certain property has a positive impact thus making each of the materials suitable for production of sustainable materials. (example: compare pumice with diatomaceous earth the starting materials for production of porous materials under hydrothermal technology; see if these materials are suitable for production of geopolymers etc.). 

Response from the authors:

We thank Reviewer 2 for this suggestion. The "Discussion" section contains comparisons with other materials reported in the literature.

Reviewer 3 Report

The manuscript contains interesting data on chemical, mineralogical, morphological, granulometric and thermal characterization of naturally expanded volcanic rocks: pumice, lapillus, zeolitic tuff and volcanic debris from Tessennano and Arlena quarry, but there are many questions for the manuscript that need to be improved. I suggest resubmitting the paper after a major revision.

1     The topic and research questions are too broad, and the research purpose should be condensed further.

2     It is undeniable that the volcanic rocks in the author's manuscript have done many test work, but for me, this manuscript is more like a test report than a scientific contribution at this moment. I think the reason is that there are too many research objects in this manuscript, and the content of the research is too superficial to go deep enough. I am not sure about the scientific contribution of this paper, since most of the results and discussions seems to be more technical than scientific.

3     Rewriting the Abstract is necessary. It ought to be succinct and provide readers with information on the background, research question, hypothesis, methodology, key findings, and conclusions of the study that is being presented. Ideally, it should also discuss the main implications and wider context of your findings.

4     I read the results and discussion section completely. In the discussion section, most of the content is about the test results or the author's speculation on the test results which means there were too few references quoted. For example, Line 370-372, “Zeolitic tuff can be exploited for various uses: if it is very fine (< 250 μm) ……”, it is necessary to add related literature here.

5     In addition, I believe that this testing result can be obtained without too much analysis. As an alternative, it is recommended to use quantitative reasoning comparing with appropriate benchmarks.

Therefore, I cannot support its publication in the journal in its present form. It should be rejected.

Author Response

Dear Editors and Reviewer of Materials

In this document, we present our response for each one of the Reviewers’ comments, as well as the list of changes and/or rebuttal made for the manuscript entitled “Characterization of volcano-sedimentary rocks and related scraps for design of sustainable materials”.

We’d like to thank the valuable comments of the Reviewers, which helped us to improve the quality of the manuscript, making it more suitable for publication in Materials than in its previous version. In the revised manuscript, all the changes have been highlighted in yellow. English language has been improved.

Reviewers' comments:

  • Reviewer #3

The manuscript contains interesting data on chemical, mineralogical, morphological, granulometric and thermal characterization of naturally expanded volcanic rocks: pumice, lapillus, zeolitic tuff and volcanic debris from Tessennano and Arlena quarry, but there are many questions for the manuscript that need to be improved. I suggest resubmitting the paper after a major revision.

1     The topic and research questions are too broad, and the research purpose should be condensed further.

2     It is undeniable that the volcanic rocks in the author's manuscript have done many test work, but for me, this manuscript is more like a test report than a scientific contribution at this moment. I think the reason is that there are too many research objects in this manuscript, and the content of the research is too superficial to go deep enough. I am not sure about the scientific contribution of this paper, since most of the results and discussions seems to be more technical than scientific.

3     Rewriting the Abstract is necessary. It ought to be succinct and provide readers with information on the background, research question, hypothesis, methodology, key findings, and conclusions of the study that is being presented. Ideally, it should also discuss the main implications and wider context of your findings.

4     I read the results and discussion section completely. In the discussion section, most of the content is about the test results or the author's speculation on the test results which means there were too few references quoted. For example, Line 370-372, “Zeolitic tuff can be exploited for various uses: if it is very fine (< 250 μm) ……”, it is necessary to add related literature here.

5     In addition, I believe that this testing result can be obtained without too much analysis. As an alternative, it is recommended to use quantitative reasoning comparing with appropriate benchmarks.

Response from the authors:

The authors thank reviewer 3. The authors wanted to analyse and report on the main work done on volcanic rocks, and materials derived from them, with the aim to create a database from which industry can take inspiration for enhancing circularity by using these raw materials in its productive cycles.

A broader context is less in-depth was purposely chosen.

The abstract has been rewritten or arranged. The discussion section was implemented with related literature.

Round 2

Reviewer 3 Report

The modifications made in the paper make it suitable for publication.